# Two-colour dissipative solitons and breathers in microresonator second-harmonic generation

Juanjuan Lu[1,4], Danila N. Puzyrev[2,3], Vladislav V. Pankratov[2,3], Dmitry V. Skryabin [2,3] ✉, Fengyan Yang[1], Zheng Gong [1], Joshua B. Surya [1] & Hong X. Tang [1] ✉

Frequency conversion of dissipative solitons associated with the generation of broadband optical frequency combs having a tooth spacing of hundreds of giga-hertz is a topical challenge holding the key to practical applications in precision spectroscopy and data processing. The work in this direction is underpinned by fundamental problems in nonlinear and quantum optics. Here, we present the dissipative two-colour bright-bright and dark-dark solitons in a quasi-phase-matched microresonator pumped for the second-harmonic generation in the near-infrared spectral range. We also found the breather states associated with the pulse front motion and collisions. The soliton regime is found to be typical in slightly phase-mismatched resonators, while the phase-matched ones reveal broader but incoherent spectra and higher-order harmonic generation. Soliton and breather effects reported here exist for the negative tilt of the resonance line, which is possible only via the dominant contribution of second-order nonlinearity.

Optical frequency comb generation in microresonators has attracted significant attention in recent years[1,2]. The key results in this area are the demonstration of temporal dissipative Kerr solitons[3] and octave-spanning combs suitable for self-referencing[4–6]. These developments have enabled a broad range of applications, such as optical clocks, coherent optical communication, exoplanet detection, and many others, see e.g.,[7–9]. Beyond becoming an outstanding test-bed for dissipative solitons, nonlinear and quantum effects in microresonators have made a profound impact in such interdisciplinary areas as pattern formation[10], synchronization of oscillators[11,12], light crystals, and topological physics in space and time[13–17].

Dissipative bright solitons and associated frequency combs in microresonators possessing Kerr nonlinearity require anomalous group-velocity dispersion (GVD) at the pump wavelength[1,3], and the dark ones are observed in the normal GVD regime[18]. Simultaneous bright and dark Kerr soliton pairs spectrally located on different sides of the zero GVD wavelength have also been recently observed[19].

Normal-dispersion Kerr resonators with the modulated circumference of the inner ring, which couples forward and backward waves, have been used to demonstrate the continuum of bright and dark dissipative Kerr soliton states[20].

The need to expand the family of microresonator combs in the visible and mid-infrared (IR) ranges stimulates interest in modelocking involving simultaneous harmonic generation, e.g., using $\chi^{(2)}$, i.e., second-order nonlinearity. $\chi^{(2)}$ effect allows generating combs at twice or half of the pump frequency at the comparatively low input powers[21–27]. An attractive feature of microresonator $\chi^{(2)}$ combs is their immediate octave width, which offers a pathway to compact self-referencing arrangements in the integrated setups[6].

Reliable generation of $\chi^{(2)}$ solitons in microresonators remains a challenge. The existence of such solitons assumes, as a necessary condition, mutual modelocking of the two groups of modes located around the pump frequency and either half- or second-harmonic[28]. One of the obstacles is, therefore, the large accumulated dispersion

[1]Department of Electrical Engineering, Yale University, New Haven, CT 06511, USA. [2]Department of Physics, University of Bath, Bath BA2 7AY, UK. [3]Centre for Photonics and Photonic Materials, University of Bath, Bath BA2 7AY, UK. [4]Present address: School of Information Science and Technology, ShanghaiTech University, Shanghai 201210, China. ✉e-mail: d.v.skryabin@bath.ac.uk; hong.tang@yale.edu

across the octave bandwidth. As a result, the group-velocity difference between the two modal groups dominates the nonlinear frequency shifts, which complicates the generation of solitons. Also, the spectral non-equidistance of neighboring mode pairs in microresonators is very substantial if compared to fiber-loop, open multi-mirror, and other types of low-repetition-rate and low-finesse resonators, where modelocking using $\chi^{(2)}$-effects have been demonstrated[29–32]. Ref. 33 provides an overview of theoretical studies of $\chi^{(2)}$ solitons in resonators between the 1990s and our days.

Lithium niobate (LN) is one of the favored $\chi^{(2)}$ materials to use in nano-fabrication for nonlinear and quantum optics applications[34]. However, LN and the other $\chi^{(2)}$ materials possess appreciable $\chi^{(3)}$, i.e., Kerr, nonlinearity. In particular, the prior soliton demonstrations in the thin-film LiNbO$_3$ resonators[35,36] have been attributed to the Kerr effect. Also, recent experiments with comb generation in AlN microresonators have revealed the strong competition between $\chi^{(2)}$ and Kerr effects[37,38]. The AlN microresonator used for the parametric down-conversion has allowed observation of bright solitons in the IR signal accompanied by the non-localized modelocked waveform in the near-IR pump[39]. Thus, the challenge of observing the two-color bright-bright or dark-dark frequency-comb solitons in the $\chi^{(2)}$-mediated high-repetition-rate microresonator frequency conversion has so far remained unresolved.

Our present work demonstrates the dissipative two-color solitons and breathers in a quasi-phase-matched microresonator pumped for second-harmonic generation in the near-IR spectral range. The soliton regime is found to be typical for phase-mismatched resonators, while phase-matched ones reveal broader but incoherent spectra and higher-order harmonic generation. Positive phase-mismatching by less than one free spectral range induces tilting of the resonance line towards negative detunings, which is possible only via the dominant contribution of second-order nonlinearity.

## Results

Here, we study the second-harmonic generation from the IR (1550 nm) to near-IR (780 nm) spectrum in the periodically polled thin-film LiNbO$_3$ microresonator. Our experimental setup is illustrated in Fig. 1. Our resonator radius is 70 μm, which provides a high repetition rate $\simeq 290$ GHz. We use the quasi-phase-matching grating to provide a large controllable positive phase mismatch, which interplays with the $\chi^{(2)}$ nonlinearity, making the resonance peak tilt towards negative detunings, see Fig. 2a.

Below, we present measurements of the optical and radio-frequency (RF) spectra, which we interpret by the existence of bright-bright and dark-dark two-color soliton pairs and breathers. We demonstrate that the bright and dark solitons merge into a single family continuously on the variation of the system parameters. The merging becomes possible through angular periodicity and small ring sizes. The blurred difference between the bright and dark solitons manifests itself in the measured and modeled periodic expansion and shrinking of the solitons. Our experiment deals with the case when the pump experiences large anomalous GVD, and the second-harmonic is in the large normal GVD range. Despite this, the pump and second-harmonic solitons have the same type, e.g., if one is bright, the other is too, which appears to be the case not yet met in the resonator and modelocking contexts. The detailed numerical analysis guides our interpretations of the data.

The width of the resonator ridge is 1.8 μm, and the vertical dimensions are 410 and 590 nm to the air–LiNbO$_3$ and LiNbO$_3$–SiO$_2$ interfaces, respectively. A bus waveguide is specifically designed for simultaneous telecom and near-visible light coupling. Following the design principle elaborated in[40], the waveguide width, wrap-around angle, and resonator-waveguide coupling gap are optimized to be 1.8 μm, 60°, and 400 nm. The resonator spectrum near the pump, $\zeta = a$, and second-harmonic, $\zeta = b$, is approximated by $\omega_{\mu\zeta} = \omega_{0\zeta} + \sum_n D_{n\zeta}\mu^n/n!$. Here, $\omega_{0a}$ and $\omega_{0b}$ are the resonator frequencies with the mode

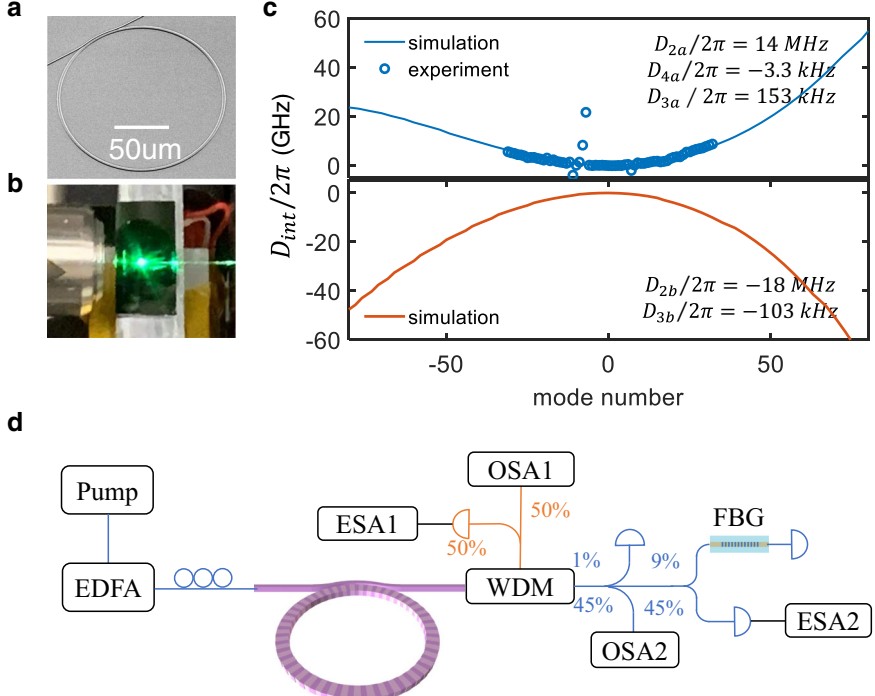

**Fig. 1 | Experimental setup and resonator dispersion. a** Scanning-electron-microscopy image of the lithium niobate microresonator. The radius of the microresonator is 70 μm, corresponding to a 286 GHz repetition rate. **b** Photograph of the chip. **c** Measured (blue circles) and computed (blue and red lines) integrated dispersion, $\omega_{\mu\zeta} - \omega_{0\zeta} - D_{1\zeta}\mu$, vs. $\mu$. Blue marks the infrared pump,

$\zeta = a$, and red marks the near-infrared second harmonic, $\zeta = b$. **d** Measurement setup: EDFA erbium-doped fiber amplifier, WDM wavelength-division multiplexer, ESA electrical spectrum analyzer, OSA optical spectrum analyzer, FBG fiber Bragg grating.

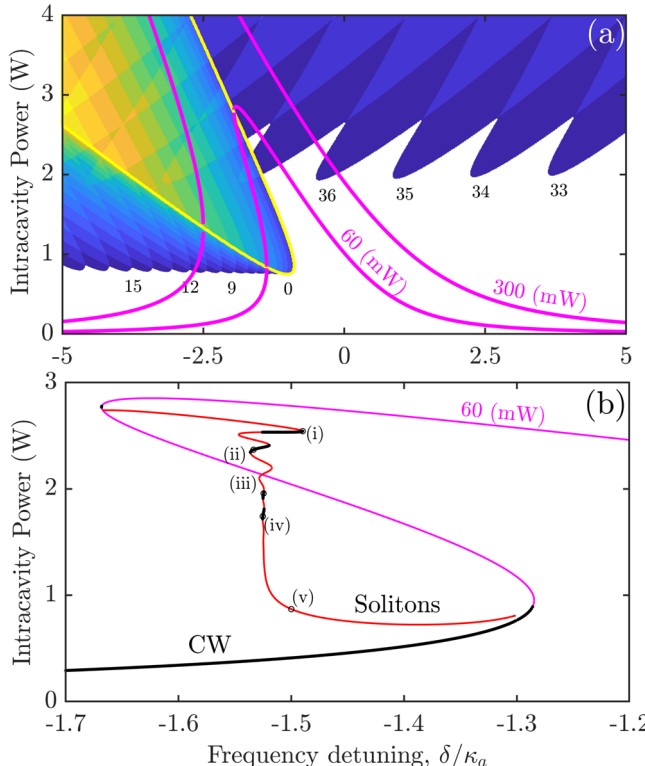

**Fig. 2 | Nonlinear single-mode solutions and soliton families. a** Magenta lines show the single-mode (continuous-wave (CW)) solutions numerically computed for the on-chip powers $\mathcal{W} = 60$ mW and 300 mW. Blue-yellow colors show the CW instability boundaries relative to the generation of the $\pm\mu$ sideband pairs. $\mu$ values are indicated where possible. **b** A family of the soliton states (red is unstable and black is stable) and CW state (black is stable and magenta is unstable) vs. detuning: $\varepsilon/2\pi = 95$ GHz, pump wavelength is 1552 nm.

numbers $M$ and $2M + Q$, respectively, where $2\pi R/Q$ is the poling period. $Q$ equals 150 in the resonator sample used in the experiments described below. $\mu = 0, \pm 1, \pm 2, \ldots$ is the relative mode number. The phase mismatch is characterized by parameter $\varepsilon$[33],

$$\varepsilon = 2\omega_{0a} - \omega_{0b}, \qquad (1)$$

which is determined by $Q$ and temperature tuning. The resonator repetition rates are $D_{1a}/2\pi = 286.24$ GHz, $D_{1b}/2\pi = 289.24$ GHz, and make the 3 GHz difference. Second-order dispersion is large anomalous near the pump, $D_{2a}/2\pi = 14$ MHz, and large normal near the second harmonic, $D_{2b}/2\pi = -18$ MHz. Linewidths are rounded to $\kappa_a/2\pi = 600$ MHz and $\kappa_b/2\pi = 1.2$ GHz. The other parameters, as well as the coupled-mode equations, are described in Methods.

The microresonator chip is placed on a piezo-positioning stage with a standard laser temperature controller set at 130 °C, limiting the variations to less than 0.01 °C and providing $\varepsilon/2\pi = 95$ GHz for the pump around 1550 nm. The pump is coupled into the microresonator using an aspheric lens (numerical aperture = 0.6) and out of waveguide with a butt-coupled fiber. The pump is amplified in an erbium-doped fiber amplifier, whose output power is set at 27 dBm. The coupling loss for the 1550 and 780 nm light is estimated to be 7–8 dB/facet and 12–13 dB/facet, respectively. The output spectra are recorded using two grating-based optical spectrum analyzers covering 350–1750 and 1500–3400 nm. The generated dual-band frequency combs are spectrally separated using a wavelength-division multiplexer. The

comb's optical and RF spectra are characterized using the grating-based optical spectrum analyzer and electrical spectrum analyzer, respectively. The resolution bandwidth of 100 kHz is utilized for the RF noise measurement. Detuning, $\delta = \omega_{0a} - \omega_p$, between the laser frequency, $\omega_p$, and the resonance at $\omega_{0a}$, is scanned from its negative (blue detuned) to the positive (red detuned) values.

The comb generation occurs when the pump frequency moves towards the resonance and makes the intra-resonator power exceed the modulational instability threshold[41]. It triggers the simultaneous growth of sidebands around the pump and its second harmonic, which further develops into the dual-band comb; see the experimental and numerical spectra in Fig. 3. To compute the regions of instabilities of the single-mode, i.e., continuous-wave (CW), operation relative to the generation of the $\pm\mu$ pairs, we apply the approximation-free part of the formalism developed in ref. 41, see Fig. 2a. The parameter space in Fig. 2a is spanned by the pump detuning $\delta$ and intra-resonator pump power in the $\mu = 0$ mode, $|a_0|^2$. The CW states are shown with the magenta lines. When the CW crosses into an instability tongue, it becomes unstable relative to the respective $\pm\mu$ mode pair. Two crossing points of the yellow ($\mu = 0$ instability) and magenta lines limit the range of the CW bistability for a given on-chip laser power, $\mathcal{W}$.

The negative direction of the tilt of the resonance curve, see Fig. 2, is determined by the $\chi^{(2)}$ effect and $\varepsilon > 0$. Hence, solitons and breathers reported below for negative detunings, $\delta < 0$, are attributed to the $\chi^{(2)}$ interaction. Kerr nonlinearity is accounted for in all our simulations, see Methods, and plays only a corrective role since the dominant $\chi^{(3)}$ effect would cause the opposite, i.e., the positive tilt of the resonance well known from the theory and observations of Kerr solitons[1,3].

The power conversion efficiency of the IR (1550 nm) frequency comb is defined as $\mathcal{W}_{IR}/\mathcal{W}$, where $\mathcal{W}_{IR}$ is the total power in all IR comb lines excluding the central one. The measured conversion increases from ~5% to ~25% as the pump detuning increases, which could be further improved by optimizing the extraction efficiency in the IR. The improvement in conversion with growing $\delta$ is evident from the measured and simulated spectra, where the central peak first dominates over the IR comb, see Fig. 3f, and then blends with it, see Fig. 3g, h.

By solving the coupled-mode equations, we have found a family of the stationary modelocked pulses, i.e., solitons, associated with the observed spectra, see Figs. 2b and 3, and determined the corresponding soliton repetition rate $\widetilde{D}_1 \neq D_1$. The soliton branch splits from the unstable high-power CW state, follows the snaking trajectory, and ends on the lower CW state. The snake line in Fig. 2b starts and ends at the points where the CW magenta line in Fig. 2a becomes unstable relative to the generation of the $\mu = \pm 1$ sidebands. Pulse profiles near the upper and lower CW states correspond to the two-color dark–dark and bright-bright solitons, respectively. The pulse profiles in the IR and NIR are practically the same, while the IR power is around one order of magnitude smaller.

The snaking soliton line in Fig. 2b corresponds to the definition of the collapsed snaking used to describe a sequence of bifurcations of dark localized structures in the Kerr models with normal dispersion[42], see also the earlier results in, e.g., ref. 43. A feature of our resonator sample is that the pulse size is comparable with the ring circumference. Therefore, periodic boundary conditions make the dark soliton transform into the bright one after several turns of the snake, see Fig. 3u−y. The above-mentioned solitons with the high conversion into the NIR comb correspond to the nearly vertical region of the snake trajectory. The duty cycles of the respective pulses in Fig. 3u−y match the measured conversion efficiency. The low (−90 dBm) levels of noise in RF spectra are characteristic of

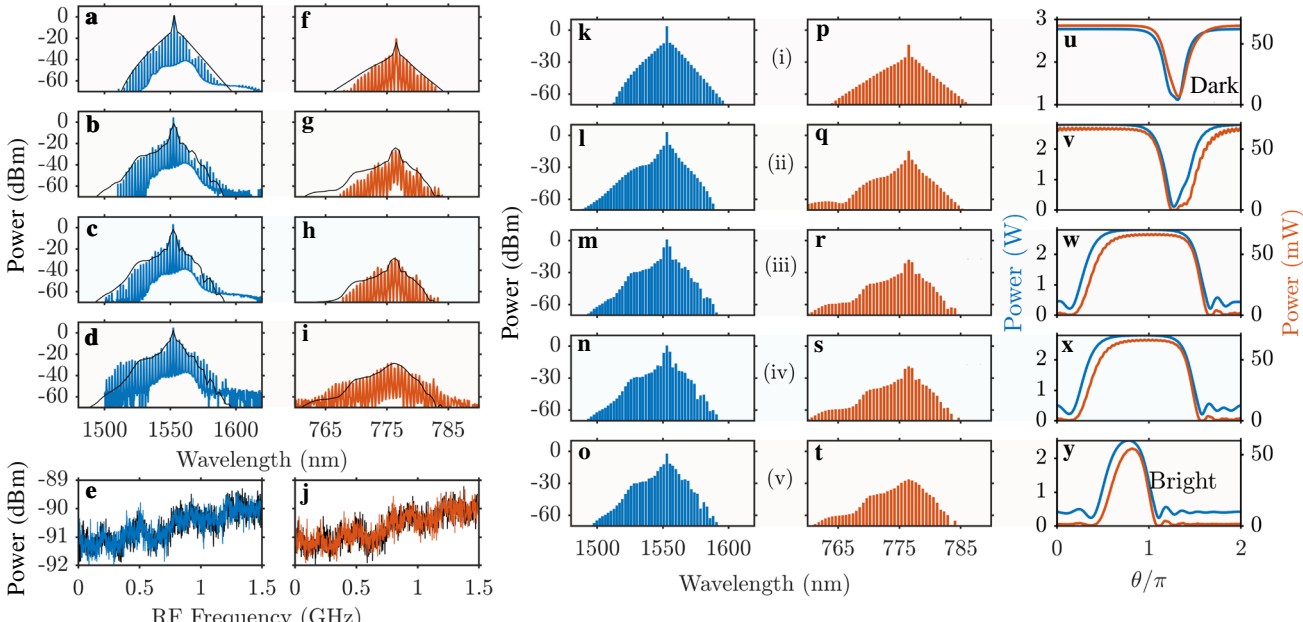

**Fig. 3 | Experimental and numerical data for soliton states.** Panels **a**–**i** show four pairs of the low-noise soliton spectra experimentally measured with the increase of the pump detuning, $\delta$. Panels **e**, **j** show the experimental RF spectra with the −90 dBm noise levels representative of the simultaneous soliton formation at 1550 and 780 nm. Black lines in **e**, **j** mark background electrical noise. Spectral envelopes shown with black lines in **a**–**d**, **f**–**i** are computed numerically and correspond to points (i)–(iv) in Fig. 2b. Panels **k**–**t** show numerical soliton spectra, and **u**–**y** show the respective pulse shapes at the points (i)–(v) marked in Fig. 2b and between the third and fourth columns here. The transition from the dark to bright solitons is clear from (**u**–**y**), which follows the collap4"n sed snaking trajectory in Fig. 2b. Left and right vertical axes in the **u**–**y** mark power for the 1550 and 780 nm pulses, respectively.

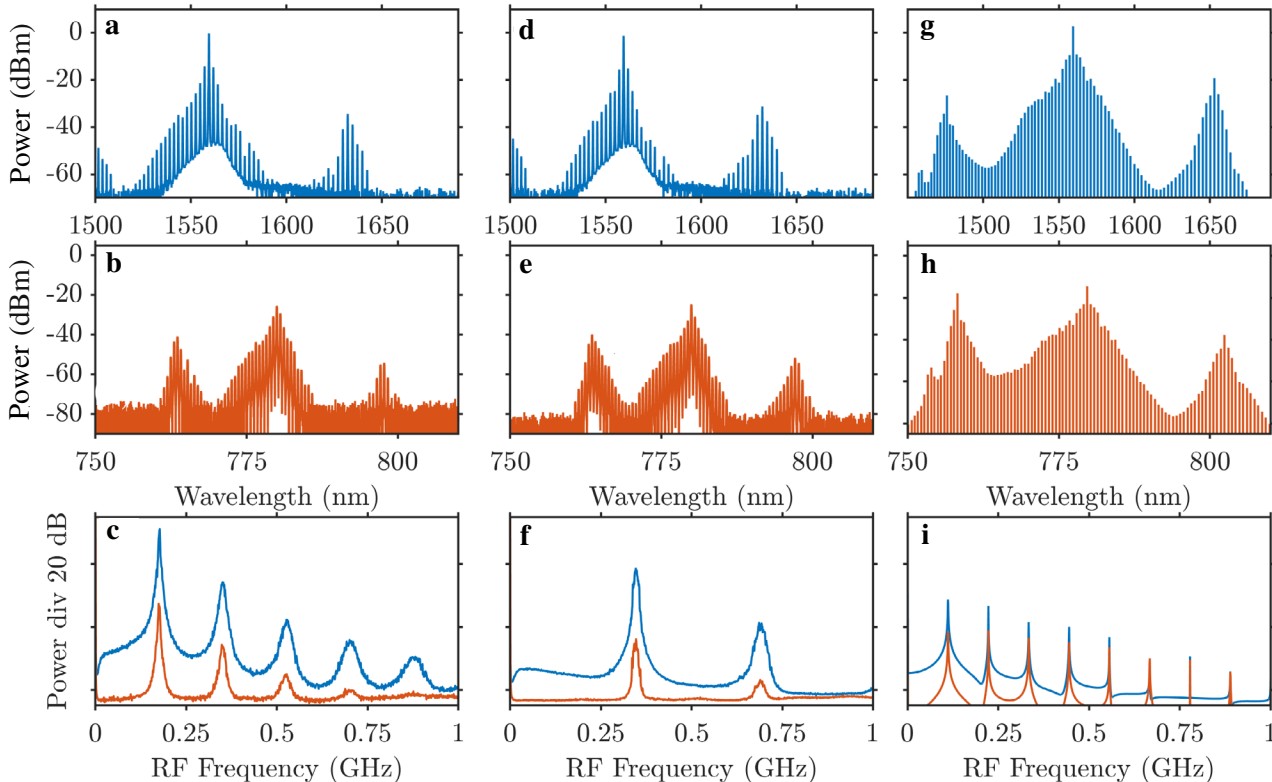

**Fig. 4 | Experimental and numerical data for breathers. a**, **b**, **d**, **e** Experimentally measured infrared and NIR spectra found on the approach to the soliton range. The corresponding RF spectra in **c**, **f** reveal the onset of modelocking via the formation of the soliton breather. Optical and RF spectra corresponding to the numerically found breathers are shown in (**g**–**i**). $\delta/\kappa_a = -1.64$, which is left from the snake turns in Fig. 2b and soliton data in Fig. 3.

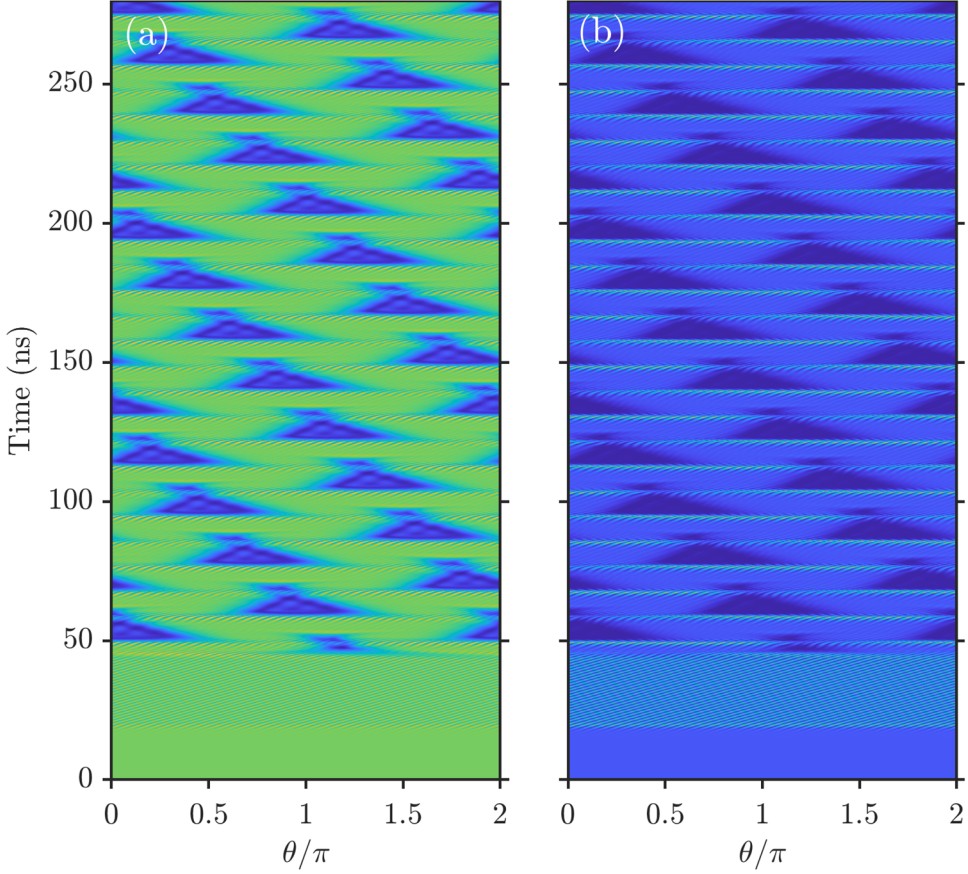

**Fig. 5 | Numerically computed breathers.** Space–time evolution of the two-color breather state, which spectra are shown in Fig. 4g–i. **a** The 1550 nm field intensity, i.e., $|A|^2$ vs. $t$ and $\theta$, and **b** The 780 nm field, $|B|^2$, see Eq. (2) in Methods.

the high degree of coherence typical for the dissipative multi-mode solitons, see Fig. 3e, j.

While doing the detuning scan and before entering the soliton regime, we first observed the characteristic three-peak spectra, see Fig. 4. The corresponding RF spectra are characterized by several well-defined peaks, see Fig. 4c–f and similar RF spectra of breathers in Kerr microresonators[44,45]. Numerical analysis reveals that such regimes correspond to the two wavefronts moving in the ring resonator. These fronts eventually meet to create a pulse, which then starts expanding again, and the cycle repeats periodically; see numerical spectra and the space–time dynamics in Fig. 4g–i and Fig. 5, respectively.

Our experiments have also revealed the trade-off between the soliton regimes for the resonator samples with large $\varepsilon$ and the generation of the high-bandwidth incoherent combs and higher order harmonics for the phase-matched resonators with $\varepsilon$ close to zero achieved by tuning the temperature to $T = 30\,°C$. The low-noise RF noise operation becomes inaccessible for $\varepsilon$ close to zero. Experimentally recorded combs in the $\varepsilon = 0$ resonator feature the broad bandwidth incoherent spectra centered around 1560 nm (pump), 780 nm (2nd harmonic), 520 nm (3rd harmonic), and 390 nm (4th harmonic), which are plotted in blue, orange, green, and purple in Fig. 6. The measured on-chip pump-to-second-harmonic-comb conversion efficiencies is around 20.7%, which is mainly limited by the bus waveguide-microring coupling condition at both near-IR and near-visible wavelength bands. The inset shows a visible light emission from the resonator captured using a CCD camera.

## Discussion

Our observations demonstrate two-colour dissipative solitons in the thin-film periodically poled LiNbO$_3$ microresonator with the ~300 GHz pulse repetition rate. A short resonator circumference limits the soliton number to one, makes possible the merging of bright and dark soliton families, and plays a role in the front-motion instability triggering breather dynamics. Solitons, breathers, and frequency comb generation reported here happen for the negative tilt of the resonance line, which is only possible via the dominant contribution of second-order nonlinearity. Future research directions include, e.g., implementing resonance tilting towards positive detunings, engineering different combinations of dispersion signs, and generation of shorter solitons and $\chi^{(2)}$ soliton crystals.

## Methods

### Device fabrication

Devices are fabricated from a commercial LN on an insulator wafer (supplied by NANOLN), in which a 590 nm thick Z-cut LN layer sits on top of 2 μm silicon dioxide on a silicon handle. The pattern is first defined using electron beam lithography (EBL) with a negative FOx-16 resist and subsequently transferred onto the LN layer using an optimized inductively coupled plasma reactive ion etching process with Ar$^+$ plasma. A thin layer of hafnium oxide is deposited on top of the fabricated photonic device using the atomic layer deposition technique, which serves as a protection layer from metal contamination induced during the poling process and also aids in confining the electric field for high-fidelity poling as a high-k material. The radial nickel electrodes are patterned on top of the LN

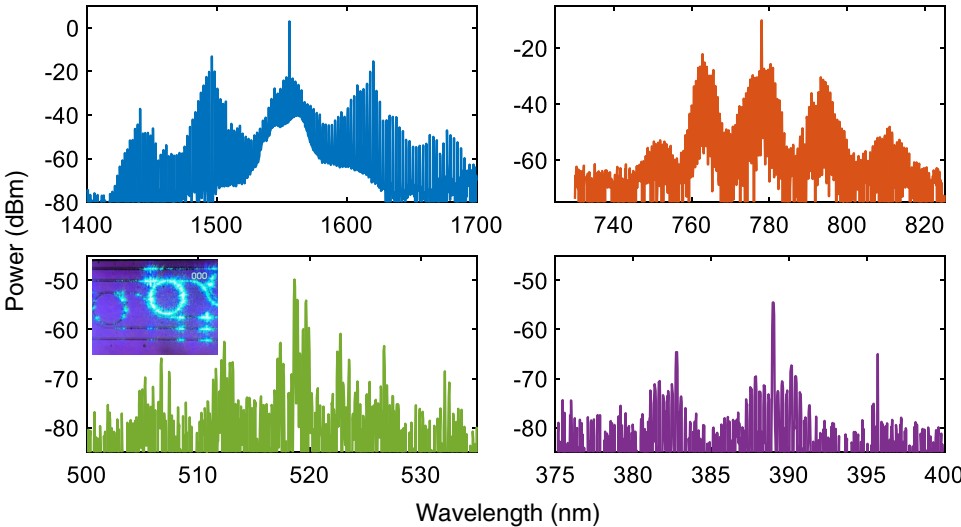

**Fig. 6 | Broadband incoherent spectra in a phase-matched microresonator.** Measurements of the broadband incoherent frequency comb generation-spanning across four octaves in the phase-matched microresonator, $\varepsilon = 0$. The on-chip pump power is $\mathcal{W} = 100$ mW.

microring concentrically via the EBL with alignment and following a bi-layer lift-off process. An optimized poling sequence was then applied to create the desired polling pattern. Afterward, the electrodes and oxide interface were sequentially removed by wet etching. Finally, the chip is cleaved to expose the waveguide facets for fiber-to-chip coupling.

## Numerical simulation

Multimode intra-resonator pump field (1552 nm, TM polarized) and it is second-harmonic (TM polarized) are expressed via their mode expansions as[33]

$$
A e^{iM\vartheta - i\omega_p t} + c.c., \quad A = \sum_\mu a_\mu(t) e^{i\mu\theta},
$$
$$
B e^{i(2M+Q)\vartheta - i2\omega_p t} + c.c., \quad B = \sum_\mu b_\mu(t) e^{i\mu\theta}, \quad (2)
$$
$$
\theta = \vartheta - \widetilde{D}_1 t.
$$

Here, $\vartheta = (0, 2\pi]$ is the angular coordinate in the laboratory frame, $\theta$ is the coordinate in the reference frame rotating with the rate $\widetilde{D}_1$, $Q = 150$ is the number of the poling periods, $M = 515$ and $\mu = 0, \pm 1, \pm 2, \dots$ is the relative mode number. $a_\mu$, $b_\mu$ are the amplitudes of the pump and second-harmonic modes.

The resonator spectrum around the pump and second harmonic is approximated as

$$
\omega_{\mu\zeta} = \omega_{0\zeta} + \mu D_{1\zeta} + \frac{1}{2!}\mu^2 D_{2\zeta} + \frac{1}{3!}\mu^3 D_{3\zeta} + \frac{1}{4!}\mu^4 D_{4\zeta}, \quad (3)
$$
$$
\zeta = a, b,
$$

where, $D_{1\zeta}$ are the linear repetition rates, $D_{2\zeta}$ are the second order dispersions, and $D_{3\zeta}$, $D_{4\zeta}$ are the third and fourth order ones. $D_{1b} - D_{1a}$ is the walk-off parameter, i.e., the repetition rate difference. The values of the dispersion coefficients are specified in Fig. 1. The pump laser, $\omega_p$, is tuned around the 1552.3 nm wavelength and targets the $\omega_{0a}$ resonance. The pump detuning is defined as

$$
\delta = \omega_{0a} - \omega_p. \quad (4)
$$

Coupled-mode equations governing the evolution of $a_\mu(t)$, $b_\mu(t)$ include both $\chi^{(2)}$ and $\chi^{(3)}$ nonlinearities[33,37]. The equations have been derived under the assumption that the $e^{iQ\vartheta}$ Fourier component of the quasi-phase-matching grating, $\chi^{(2)} G(\vartheta) = \chi^{(2)} G(\vartheta + 2\pi/Q)$, provides the

required phase-matching[33],

$$
i\partial_t a_\mu = \delta_{\mu a} a_\mu - \frac{i\kappa_a}{2}\left(a_\mu - \widehat{\delta}_{\mu,0}\mathcal{H}\right)
$$
$$
- \gamma_{2a} \sum_{\mu_1\mu_2} \widehat{\delta}_{\mu,\mu_1-\mu_2} b_{\mu_1} a^*_{\mu_2}
$$
$$
- \gamma_{3a} \sum_{\mu_1\mu_2\mu_3} \widehat{\delta}_{\mu,\mu_1+\mu_2-\mu_3} a_{\mu_1} a_{\mu_2} a^*_{\mu_3}
$$
$$
- 2\gamma_{3a} \sum_{\mu_1\mu_2\mu_3} \widehat{\delta}_{\mu,\mu_1+\mu_2-\mu_3} a_{\mu_1} b_{\mu_2} b^*_{\mu_3},
$$
$$
i\partial_t b_\mu = \delta_{\mu b} b_\mu - \frac{i\kappa_b}{2} b_\mu \quad (5)
$$
$$
- \gamma_{2b} \sum_{\mu_1\mu_2} \widehat{\delta}_{\mu,\mu_1+\mu_2} a_{\mu_1} a_{\mu_2}
$$
$$
- \gamma_{3b} \sum_{\mu_1\mu_2\mu_3} \widehat{\delta}_{\mu,\mu_1+\mu_2-\mu_3} b_{\mu_1} b_{\mu_2} b^*_{\mu_3}
$$
$$
- 2\gamma_{3b} \sum_{\mu_1\mu_2\mu_3} \widehat{\delta}_{\mu,\mu_1+\mu_2-\mu_3} b_{\mu_1} a_{\mu_2} a^*_{\mu_3}.
$$

Here, $\widehat{\delta}_{\mu,\mu'} = 1$ for $\mu = \mu'$ and is zero otherwise. $\mathcal{H}$ is the pump parameter, $\mathcal{H}^2 = \eta\mathcal{F}\mathcal{W}/2\pi$, where $\mathcal{W}$ is the laser power, and $\mathcal{F} = D_{1a}/\kappa_a$ is finesse[33]. $\eta$ is the coupling coefficient, which was used as the fitting parameter, $\eta = 0.33333$. $\delta_{\mu\zeta}$ are the modal detuning parameters in the rotating reference frame,

$$
\delta_{\mu a} = (\omega_{\mu a} - \omega_p) - \mu\widetilde{D}_1,
$$
$$
\delta_{\mu b} = (\omega_{\mu b} - 2\omega_p) - \mu\widetilde{D}_1, \quad (6)
$$

where $\delta_{0a} = \delta$, $\delta_{0b} = 2\delta - \varepsilon$ and $\varepsilon$ is the phase mismatch parameter[33],

$$
\varepsilon = 2\omega_{0a} - \omega_{0b} = 2\frac{cM}{Rn_M} - \frac{c(2M+Q)}{Rn_{2M+Q}}. \quad (7)
$$

Here, $c$ is the vacuum speed of light, $R$ is the resonator radius, $n_M$ is the effective refractive index experienced by the resonator mode with the number $M$. The value of $\varepsilon$ can be controlled by the temperature and pump wavelength. $T = 130\,^\circ$C yields $\varepsilon/2\pi \approx 95$ GHz and was set to get the soliton and breather generation in Figs. 2–4. $T = 30\,^\circ$C gave $\varepsilon \approx 0$ and was used to generate the incoherent multi-octave spectra in Fig. 5.

$\gamma_{2\zeta}$ and $\gamma_{3\zeta}$ are parameters specifying the strength of the second and third-order nonlinear effects[33]. Using a simplifying assumption that the effective mode area, $S$, does not disperse, we estimate $\gamma_{2\zeta}$ as

$$\gamma_{2\zeta} = \frac{d\omega_{0\zeta}q}{3n_0^2}, q = \sqrt{\frac{2\mathcal{Z}_{vac}}{Sn_0}}, \zeta = a,b, \qquad (8)$$

where $n_0 = 2.2$ is the linear refractive index, $\omega_{0a}/2\pi = 193\,\text{THz}$, $\omega_{0b}/2\pi = 386\,\text{THz}$, $\mathcal{Z}_{vac} = 1/\epsilon_{vac}c = 377\,V^2/W$ is the free space impedance, and the averaged effective area $S \approx 1.5\,\mu\text{m}^2$. These values yield $q \approx 15 \times 10^6\,\text{W}^{-1/2}\,\text{V/m}\,d \sim \chi^{(2)}$ is the relevant element of the reduced $\chi^{(2)}$ tensor, $d \approx 20\,\text{pm/V}$ and $\gamma_{2a}/2\pi \approx 4\,\text{GHz}/\sqrt{\text{W}}, \gamma_{2b}/2\pi \approx 8\,\text{GHz}/\sqrt{\text{W}}$. Kerr parameters $\gamma_{3\zeta}$ are estimated using the results derived in ref. [46]

$$\gamma_{3\zeta} = \frac{\omega_{0\zeta}n_2}{2Sn_0}, \qquad (9)$$

where $n_2 = 9 \times 10^{-20}\,\text{m}^2/\text{W}^2$ ($\chi^{(3)} = 1.6 \times 10^{-21}\,\text{m}^2/\text{V}^2$) is the Kerr coefficient of LiNbO$_3$ and $\gamma_{3a}/2\pi \approx 2.5\,\text{MHz/W}$ and $\gamma_{3b}/2\pi \approx 5\,\text{MHz/W}$.

According to the estimates based on the comparison of the nonlinear resonance shifts induced by the $\chi^{(2)}$ and $\chi^{(3)}$ terms[41], the latter is expected to play a notable role in $|\varepsilon|$ becoming close to and exceeding $\varepsilon_{cr}$,

$$\varepsilon_{cr} = \gamma_{2a}\gamma_{2b}/\gamma_{3a} \approx 2\pi \times 2\,\text{THz}, \qquad (10)$$

which corresponds to the mismatch by about six modes in a resonator with a 300 GHz repetition rate and is much larger than $\varepsilon/2\pi \simeq 0.1\,\text{THz}$ in our resonator.

Typical time-dependent simulations of Eq. (5) were performed using the fourth-order Runge–Kutta method applying $\widetilde{D}_1 = D_{1a}$. Stationary soliton profiles were found using the Newton method after $\partial_t$ was set to zero and $\widetilde{D}_1$ assumed as one of the unknowns. The value of $\widetilde{D}_1$ after calculations was close, but not equal, to $D_{1a}$. A typical number of modes around $\omega_p$ and $2\omega_p$ used in the modeling was 256, i.e., $\mu = -127, ..., 0, ..., 128$.

To differentiate between stable and unstable solitons, we have analyzed the linear stability of the soliton family. We perturbed the time-independent soliton amplitudes, $\hat{a}_\mu$, $\hat{b}_\mu$ ($\partial_t\hat{a}_\mu = \partial_t\hat{b}_\mu = 0$) with small perturbations, $\varepsilon_{a\mu}(t) = x_{a\mu}(t) + y_{a\mu}^*(t)$ and $\varepsilon_{b\mu}(t) = x_{b\mu}(t) + y_{b\mu}^*(t)$, i.e.,

$$\begin{aligned} a_\mu(t) &= \hat{a}_\mu + x_{a\mu}(t) + y_{a\mu}^*(t), \\ b_\mu(t) &= \hat{b}_\mu + x_{b\mu}(t) + y_{b\mu}^*(t). \end{aligned} \qquad (11)$$

and then linearized Eq. (5). By substituting $\{x_{a\mu}(t), y_{a\mu}(t), x_{b\mu}(t), y_{b\mu}(t)\} = \{X_{a\mu}, Y_{a\mu}, X_{b\mu}, Y_{b\mu}\}e^{\lambda t}$, we have reduced the linearized differential equations to the algebraic $(4 \times 256) \times (4 \times 256)$ eigenvalue problem, which was solved numerically[41]. The soliton is stable if all $\text{Re}\lambda < 0$. The stability of the CW state, $a_{\mu \neq 0} = 0$, $b_{\mu \neq 0} = 0$, was found from the simpler four-by-four eigenvalue problem[41]. In this case, each eigenvalue $\lambda$ is attributed to a particular pair of $\pm\mu$ sidebands producing its own instability boundary, which are all plotted in Fig. 2b.

## Data availability
The data supporting the findings of this study are available from corresponding authors on reasonable request.

## Code availability
The codes for data processing are available from authors on reasonable request.

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

## Acknowledgements

This work was supported in part by the NSF Center for Quantum Networks under grant number EEC-1941583, EU Horizon-2020 MSCA program (812818), and the Royal Society (SIF/R2/222029). We acknowledge partial funding support for materials used in this project by the Department of Energy, Office of Basic Energy Sciences, Division of Materials Sciences and Engineering under Grant DE-SC0019406. The authors thank M. Rooks and Y. Sun for assistance in device fabrication.

## Author contributions

J.L. performed the device design, fabrication, and measurement with the assistance of F.Y., Z.G., and J.B.S. D.N.P., V.V.P., and D.V.S. developed theory and performed numerical simulations. J.L. and D.V.S. wrote the paper text. All authors commented on the paper. D.V.S. and H.X.T. conceptualized and supervised the project.

## Competing interests

The authors declare no competing interests.
