## [Peer review file · Nature Communications]

REVIEWER COMMENTS

Reviewer #1 (Remarks to the Author):

The authors demonstrate bright-bright and dark-dark solitons based on second-order nonlinearity of thin-film lithium niobate ring resonator. The authors studied it theoretically and experimentally. The X(2) soliton is known for years and has been demonstrated on AIN platform, in the past. This time author achieved it in the LN platform and generated both bright and dark solitons in a single device. The result here would be of interest to a broad audience. I would like to recommend acceptance of the manuscript for publication in Nature Communications given that the following questions/comments can be addressed.

In general, the (2) soliton is hard to achieve octave-spanning, due to the dispersion engineering requirement in both NIR and visible range. Is there any real application that can be achieved by the author's result here?

In the author's previous LN Kerr soliton paper, the photorefractive effect plays an important and strong role. But author didn't include this effect in their simulation.

There are some parts of the experiment haven't been presented very clearly:

What kind of LN wafer did the author use? X cut? Z cut? Or Y cut?

The RF spectra in Figure 3 is for 1550 comb or 780 comb? Author should measure RF spectra for both wavelength ranges.

For figure3 (e), which color is for which point?

What's the detailed design for the bus waveguide?

The coupling loss for both 1550 nm and 780nm is 7-8 dB / facet?

Why does the conversion efficiency of the incoherent frequency comb seem higher than that of the soliton?

Reviewer #2 (Remarks to the Author):

In this paper, the authors experimentally and numerically achieved the bright-bright, dark-dark solitons in the thin-film periodically polled LiNBO₃ micro resonator through second harmonic generation. They first showed the two-color soliton spectra with various pump detuning and found that consistent with the numerical simulation. Then they demonstrated the soliton breather state through the RF spectra, which matches well with the simulation. In the end, they showed the broader and incoherent higher order comb generation for the phase matched resonator.

This paper solves an important question in the field by realizing the bright-bright, dark-dark soliton formation in the χ^2 high repetition rate micro resonator. I would recommend it for publication after the following issues have been addressed.

1. The details on the device itself are lacking. It would be much clearer to the general audience if the authors can illustrate more on their device details such as the device schematics, etc
2. In Fig.3 b, c, d, g, h, i, the measurement data does not match the simulation well. The authors need to include this in the discussion.
3. The authors had emphasized that Kerr nonlinearity does not play an important role in the soliton formation. It would be more persuasive to include direct evidence to the readers. For example, a visualization with and without χ^3 terms and an illustration when χ^3 does become important. It is briefly touched in the end of the "Methods", but it is worth to be extended. If article length is an issue, it can go into the supplementary.
4. In the Methods session, it is all about theoretical derivations. The authors need to include more experimental methods such as fabrication process etc.

Some minor issues:

1. Fig 4 g: the x-axis is different than that of other figures. It starts from 1450nm instead of 1500nm.
2. The format of references is not consistent, i.e. ref 8, ref9 have different citation format.

Reviewer #3 (Remarks to the Author):

In the paper 'Bright and dark frequency combs solitons in microresonator second-harmonic generation', the authors studied the two-color frequency combs in a quasi-phase-matched lithium niobate

microresonator. They claim the observed frequency combs at both 1550 nm band and 780 nm band correspond to two synchronized dissipative solitons in temporal domain, which can be solely attributed to second-order nonlinearity with negligible impact from Kerr nonlinearity.

The topic of dissipative cavity soliton based on second-order nonlinearity is quite new and has attracted more and more attention. In terms of scientific importance, I think this work is interesting and novel. My main concern of this work is that the experimental demonstration of two-color solitons is inadequate. The optical spectra and RF noise measurement can only illustrate the combs are coherent, which cannot provide complete temporal pulse information, not mentioning if the pulses are in synchronization. Indeed, there are some indirect evidence such as the optical spectra as a function of the detuning as well as the numerical simulations, however, without temporal characterization (either direct measurement or indirect methods), I do not get convinced with the main conclusion- observation of two-color solitons. Therefore, I cannot support the current paper for publication in Nature communications.

Other questions and comments:

1. What is the fundamental difference between the 'bright soliton' and 'dark soliton', is there a proper way to distinguish them? It would be more accurate to call them mode-locked pulse.
2. Are the RF spectra measured for 1550 nm band or 780 nm band?

Dear Reviewers,

We greatly appreciate your thorough checking of our manuscript. We revised the manuscript per your comments and provided our detailed response below. We apologize for the longer-than-usual period of revisions due to the need to repeat and extend some of the measurements, staff finishing their contracts, illnesses and pressures on lab time.

Sincerely Yours,
Authors

Reviewer #1

1) The authors demonstrate bright-bright and dark-dark solitons based on second-order nonlinearity of thin-film lithium niobate ring resonator. The authors studied it theoretically and experimentally. The X(2) soliton is known for years and has been demonstrated on AIN platform, in the past. This time author achieved it in the LN platform and generated both bright and dark solitons in a single device. The result here would be of interest to a broad audience. I would like to recommend acceptance of the manuscript for publication in Nature Communications given that the following questions/comments can be addressed.

Reply: We thank the Reviewer's positive comments and recommendation for accepting after revisions.

2) In general, the (2) soliton is hard to achieve octave-spanning, due to the dispersion engineering requirement in both NIR and visible range. Is there any real application that can be achieved by the author's result here?

Reply: It is challenging for chi² soliton to have broad continuous spectra because nonlinear response depends on the phase-matching bandwidth, which is typically narrow. On the other hand, their two spectral parts are separated by an octave wide. Their main application is the frequency-comb generation across the wavelength ranges that cannot be easily accessed by commercial pump lasers (infrared) or across the spectral ranges having unfavourable (normal) dispersion, i.e. visible and ultraviolet, see lines 45-54.

3) In the author's previous LN Kerr soliton paper, the photorefractive effect plays an important and strong role. But author didn't include this effect in their simulation.

Reply: In this work, TM mode of Z cut LN (extraordinary guided mode) is used for the comb generation, where the thermal-optic effect dominates the process. However, in our previous LN Kerr soliton paper, TE mode in Z cut LN is used instead, which corresponds to the ordinary guided modes.

We note that LN exhibits a significant thermo-optic effect for extraordinary light ($\frac{dn_e}{dT} \approx 3.34 \times 10^{-5} \text{K}^{-1}$), while it is negligible for ordinary light ($\frac{dn_o}{dT} \approx 0$) around room temperature (see L. Moretti, M. Iodice, F. G. D. Corte, and I. Rendina, “Temperature dependence of the thermo-optic coefficient of lithium niobate, from 300 to 515 K in the visible and infrared regions,” J. Appl. Phys. 98, 036101 (2005)). Accordingly, in the experiment, we found that the photorefractive effect plays an important role in the scenario where ordinary light is used, while it is negligible in this work where extraordinary light is used. Also, the photorefractive effect is too slow to impact the established soliton regime associated with ps pulses.

4) *There are some parts of the experiment haven't been presented very clearly: What kind of LN wafer did the author use? X cut? Z cut? Or Y cut?*

Reply: Z cut lithium-niobate-on-insulator wafer is used. We have added more information regarding the material and device fabrication in the Methods section.

5) *The RF spectra in Figure 3 is for 1550 comb or 780 comb? Author should measure RF spectra for both wavelength ranges.*

Reply: We have done so now and found that RF spectra for the 1550nm and 780nm combs are both below -80dBm. Two RF measurements are shown separately in the figure S1

Fig. S1. Representative $\chi^{(2)}$ soliton microcomb spectra at both 1550nm and 780nm wavelength bands and their corresponding RF spectra.

above. Both measurements unambiguously confirm simultaneous, i.e., dual-band, low-noise modelocking regime. Furthermore, we have also performed simultaneous 1550nm

and 780nm RF measurements in the breather regime. These measurements confirmed the complete correlation of the breathing dynamics between the pump and second harmonic pulses. The corresponding data are included in the revised Fig. 4. Thus, the dual-band RF measurements confirm our conclusions about forming the two-color solitons and breathers.

6) For figure 3 (e), which color is for which point?

Reply: The RF spectra shown in 3(e) (1550nm) and 3(j) (780nm) are practically the same for all four optical spectra shown above.

7) What's the detailed design for the bus waveguide?

Reply: A complete information on this is now included in Lines 120-125.

8) The coupling loss for both 1550 nm and 780nm is 7-8 dB / facet?

Reply: Coupling loss for the 1550nm and 780nm are around 7-8 dB/facet and 12-13 dB/facet, respectively. In the experiment, a cleaved single-mode fiber with a core diameter of 9 μ m is utilized for butt-coupling 1550nm and 780nm light from the bus waveguide. Since visible light suffers more from the facet roughness and mode mismatch, it has a slightly larger coupling loss compared to that of 1550nm light. To clarify, we have modified the sentence accordingly in the revised text, see Line 154.

9) Why does the conversion efficiency of the incoherent frequency comb seem higher than that of the soliton?

Reply: Close to the frequency matching (ϵ near zero) condition, it is much more challenging to achieve the soliton state while the conversion efficiency is high. However, under large $|\epsilon|$, the conversion efficiency is compromised while soliton formation is favored through the sum/difference frequency cascades, which lock the phases of interacting modes (Puzyrev et al, Physical Review A 104, 013520 (2021)).

Reviewer #2

1) In this paper, the authors experimentally and numerically achieved the bright-bright, dark-dark solitons in the thin-film periodically polled LiNBO3 micro resonator through second harmonic generation. They first showed the two-color soliton spectra with various pump detuning and found that consistent with the numerical simulation. Then they demonstrated the soliton breather state through the RF spectra, which matches well with the simulation. In the end, they showed the broader and incoherent higher order comb generation for the phase matched resonator.

This paper solves an important question in the field by realizing the bright-bright, dark-dark soliton formation in the χ^2 high repetition rate micro resonator. I would recommend it for

publication after the following issues have been addressed.

Reply: We thank the Reviewer's positive comments and recommendation for publication after revisions.

2) The details on the device itself are lacking. It would be much clearer to the general audience if the authors can illustrate more on their device details such as the device schematics, etc

Reply: As suggested by the Reviewer, we have added more details regarding the device parameters in the main text and fabrication process in the Methods section of the revised manuscript, see Lines 118-125, 278-299.

3) In Fig.3 b, c, d, g, h, i, the measurement data does not match the simulation well. The authors need to include this in the discussion.

Reply: We note that the number of parameters we use in our model is more than twice relative to the more established for the theory-experiment comparison Kerr resonators. Therefore, scanning through the parameter space to achieve a perfect agreement is counterproductive and not necessary. We are pleased about our agreement between our modelling and experiment, especially regarding the observed sequences of the spectra and their shapes, and the discrepancies the referee notes in Fig. 3 do not affect our claims.

4) The authors had emphasized that Kerr nonlinearity does not play an important role in the soliton formation. It would be more persuasive to include direct evidence to the readers. For example, a visualization with and without χ^3 terms and an illustration when χ^3 does become important. It is briefly touched in the end of the "Methods", but it is worth to be extended. If article length is an issue, it can go into the supplementary.

Reply: The explicit argument showing that we are dealing with the dominant χ^2 effect has now been spelt out across the text (see Lines 17, 99, 180-188, 270-274), and was, in fact, already contained in the original version in Fig. 2. Indeed, Kerr nonlinearity acting on its own would provide a positive tilt of the resonance lines in Fig. 2, i.e., bistability range would happen for positive detunings. The negative direction of the tilt of the resonance curves in Fig. 2, is determined by the χ^2 effect and $\epsilon > 0$. Hence, solitons and breathers reported below for negative detunings, $\delta < 0$, are attributed to the χ^2 interaction. Kerr nonlinearity is included in all our simulations and plays only a corrective role in reshaping the bistability curve, which remains firmly in the negative detuning range. The dominant χ^3 effect would cause the opposite, i.e., positive tilt of the resonance well known from the theory and observations of Kerr solitons. Distinguishing χ^2 and χ^3 effects become more subtle for $\epsilon < 0$, when both χ^2 and χ^3 tilts are positive, but this is not the case in our resonator.

5) In the Methods session, it is all about theoretical derivations. The authors need to include more experimental methods such as fabrication process etc.

Reply: As the reviewer suggested, we have added more information regarding the device parameters in the revised main text and a subsection entitled device fabrication in the Methods section.

6) *Some minor issues:*
Fig 4 g: the x-axis is different than that of other figures. It starts from 1450nm instead of 1500nm.

Reply: We thank the reviewer for pointing this out. Figs.4a and 4d are the experimental data recorded using the optical spectrum analyzer (AQ6376E) covering 1500–3400 nm. Fig. 4g is the simulated spectrum, which shows more comb features in the wavelength <1500nm and matches well with the experimental data (Fig.4d).

7) *The format of references is not consistent, i.e. ref 8, ref 9 have different citation format.*

Reply: We made them consistent.

Reviewer #3

1) In the paper 'Bright and dark frequency combs solitons in microresonator second-harmonic generation', the authors studied the two-color frequency combs in a quasi-phase-matched lithium niobate microresonator. They claim the observed frequency combs at both 1550 nm band and 780 nm band correspond to two synchronized dissipative solitons in temporal domain, which can be solely attributed to second-order nonlinearity with negligible impact from Kerr nonlinearity. The topic of dissipative cavity soliton based on second-order nonlinearity is quite new and has attracted more and more attention. In terms of scientific importance, I think this work is interesting and novel. My main concern of this work is that the experimental demonstration of two-color solitons is inadequate. The optical spectra and RF noise measurement can only illustrate the combs are coherent, which cannot provide complete temporal pulse information, not mentioning if the pulses are in synchronization. Indeed, there are some indirect evidence such as the optical spectra as a function of the detuning as well as the numerical simulations, however, without temporal characterization (either direct measurement or indirect methods), I do not get convinced with the main conclusion- observation of two-color solitons. Therefore, I cannot support the current paper for publication in Nature communications.

Reply: We appreciate the Reviewer's comments and feedbacks for our work. We agree with the reviewer that *"The optical spectra and RF noise measurement can only illustrate the combs are coherent, which cannot provide complete temporal pulse information."* As recommended by the reviewer, we have attempted to use the autocorrelator to directly measure the temporal information of the generated mode-locked combs. Unfortunately, the autocorrelator (IR GRENOUILLES 15-100-USB) we have could only measure pulse length range of 70fs-1.5ps while the solitons have a width of > 1.5 ps as inferred from the simulation result (round-trip time is ~3.5ps). As a result, we could not verify their temporal shape directly. However, the measured comb shapes, newly done dual-band RF measurements, and our simulation data, taken either independently or in combination,

unambiguously support the observation of two-color solitons and breathers. Although we could not directly measure pulse durations, this would only be another addition to the existing evidence supporting our claim of the two-color-solitons.

2) *Other questions and comments: What is the fundamental difference between the 'bright soliton' and 'dark soliton', is there a proper way to distinguish them? It would be more accurate to call them mode-locked pulse.*

Reply: The difference between the bright and dark pulses is blurred in a ring geometry. Fig. 3 and Lines 215-230 elucidate the interplay and merger of the bright and dark pulses into the same family of pulses. All “solitons” reported in dozens of microresonators are, in fact, trains of modelocked pulses with discrete spectra. The established tradition now is to call them solitons, which is justified in our opinion also.

3) *Are the RF spectra measured for 1550 nm band or 780 nm band?*

Reply: We have now measured RF spectra for 1550nm and 780nm bands. Demonstrating simultaneous -90 dBm noise levels and well-correlated breather peaks leave no doubt that the corresponding multi-mode spectra are associated with modelocked pulses, traditionally called by the microresonator community - solitons. The measured soliton spectra for both spectral bands are shown in section 5) of our response to the referee one, Figs 3e, 3j, and the correlated RF peaks corresponding to the breather spectra are included in the revised Fig. 4. The new RF measurements, also invited by other referees, support the claim that the modelocking happens in both spectral bands simultaneously.

REVIEWERS' COMMENTS

Reviewer #1 (Remarks to the Author):

The authors answered my questions and I don't have more comments about this manuscript.

Reviewer #2 (Remarks to the Author):

The authors have addressed all my concerns. I now recommend it for publication on Nature Communications. I appreciate the work and efforts that the authors put into this revision.

Reviewer #3 (Remarks to the Author):

The revised manuscript includes RF spectra measurement for both 780 nm and 1550 nm bands, which is critical indirect evidence for claiming of the observation of two-color mode locking. As I said before, the observation of two mode-locked pulses linking through $\chi^{(2)}$ in a microresonator is quite interesting. Since the authors provide new solid indirect evidence, I get convinced with their claim now. So, I would recommend the revised manuscript for publication in Nature Communications after minor changes.

1. Please show the RF noise floor in Fig. 3(e) and (j).
2. Please give a brief description of RF spectra measurement setup?

We greatly appreciate latest comments and recommendations to accept and publish, in principle, a suitably revised version in Nature Communications. Our response to the comments and implemented revisions are listed below.

Sincerely Yours,
Authors

Reviewer 3 comment: Please show the RF noise floor in Fig. 3(e) and (j).

Reply: RF spectra of the pump and 2nd harmonic in Figs. 3e and 3j in the previous manuscript were measured at different resolution bandwidths (RBWs). We remeasured them with the same spectrum analyzer (RBW=100kHz) to ensure consistency. We updated data in Figs. 3e and 3j, accordingly, and also included the respective background electrical noise, see black lines.

Reviewer 3 comment: Please give a brief description of RF spectra measurement setup?

Reply: To address this comment, we added a schematic measurement setup in Fig. 1d and its description in Lines 159-164.

Other changes:

Throughout previous versions and on top of the soliton results, the manuscript abstract, summary, a part of its text, references and displayed data (Figs. 4 and 5) also described observations of the breather states. To reflect on this, we have incorporated the “breather” term into the title so that now it reads as “Frequency-comb solitons and breathers...”